# The Sound of Silence: Unspoken Meaning in the Discourse of Pregnant and Breastfeeding Women on Environmental Risks and Food Safety in Spain

**DOI:** 10.3390/nu14030593

**Published:** 2022-01-29

**Authors:** Miguel Company-Morales, Lina Casadó, Eva Zafra Aparici, María Filomena Rubio Jiménez, Andrés Fontalba-Navas

**Affiliations:** 1Seron Primary Care Center, Northern Almería Integrated Healthcare Area, 04600 Huercal-Overa, Almeria, Spain; marienarubiojimenez@gmail.com; 2Department of Nursing, Physiotherapy and Medicine, University of Almería, 04120 La Cañada, Almeria, Spain; 3Department of Nursing, Medical Anthropology Research Centre (MARC), University Rovira i Virgili, 43003 Tarragona, Tarragona, Spain; linacristina.casado@urv.cat; 4Department of Anthropology, Philosophy and Social Work, University Rovira i Virgili, 43003 Tarragona, Tarragona, Spain; eva.zafra@urv.cat; 5Antequera Hospital, Northern Málaga Integrated Healthcare Area, 29200 Antequera, Malaga, Spain; andres.fontalba.sspa@juntadeandalucia.es; 6Department of Public Health and Psychiatry, University of Málaga, 29016 Málaga, Malaga, Spain

**Keywords:** food, pregnancy, breastfeeding, environmental pollution, silences, biopolitics, learned helplessness

## Abstract

(1) Background: In recent years, a growing number of qualitative health research studies have performed discourse analysis of data from participants’ narratives. However, little attention has been paid to the gaps and silences within these narratives. The aim of the present study is to interpret the silences detected in the discourse of pregnant and breastfeeding women concerning environmental risks and food safety. (2) Methods: This descriptive, interpretive, observational study was conducted according to a qualitative research paradigm and from a phenomenological and ethnographic perspective. The study sample was composed of 88 intentionally selected women, among whom fifty interviews, three ethnographies and five focus groups were conducted. Data coding and analysis were performed using N-Vivo 12 software. (3) Results: The results obtained show that the women’s discourse presented silences that reflected their minimisation of perceived environmental and food risks. However, these women were wary of food produced in the proximity of contaminated areas. Nevertheless, the participants believed they were powerless to overcome environmental pollution and the potential contamination of their own bodies. (4) Conclusions: The participants’ minimisation of the environmental risks faced and their inaction in this respect are sustained by the biopolitical practices of public institutions, which have propelled these women into a situation of learned helplessness and social injustice.

## 1. Introduction

Silence is a meaningful resource that adds light and shade to communicative practices and merits the attention of qualitative researchers. People communicate their ideas, values and beliefs mainly via the spoken word and non-verbal language but also through their silence. Studies have observed the importance of silence in their interpretation of discourse, in the view that “what is left unsaid” is also of pragmatic value [1,2]. This pragmatic meaning of silence during a conversation may or may not be captured and interpreted by those addressed [3,4]. Moreover, if it is not to be diffuse and valueless, such a silence must always be interpreted in relation to the prevailing context and the dialogue taking place. Thus, in conjunction with relationships among social norms, silence highlights the overlapping of diverse contexts. For the qualitative researcher, it is an essential element for the interpretation of discourse [5], and it has been argued that contextualised silence reflects at least one meaning in each communicative situation [1,6].

Social theorists have argued that silence during discourse is an affirmation of non-existence, and hence the verification that there is nothing to say, or to see, or to know, on the subject under discussion [7,8]. Some silences, moreover, can be interpreted as a passive, powerless and cautious attitude that is subjectively adopted when people believe they cannot influence a given situation [9,10,11]. Seligman (1985) called this behaviour “learned helplessness”, and defined it as an actionless response to a stimulus viewed as uncontrollable. People who generate learned helplessness believe that no action of theirs will have any appreciable effect, and so they do not react. According to the theory of universal defencelessness, a similar phenomenon can be observed at the social level, when a population group fails to react to a stimulus. This understanding explains some aspects of social passivity. In macroscopic terms, each group member considers the situation uncontrollable; neither they nor their peers can sway the outcome in the direction desired by the group [12,13].

Camargo and Méndez [2,14,15] have studied the pragmatic functions of silence in Spanish society, analysing the situational contexts in which they occur. For these authors, silence has various meanings, with diffuse borders, that might be situated within a continuum built according to the Prototype Theory [16]. Thus, four types of silences were identified, with the following pragmatic functions: discursive, structured, epistemological/psychological and normative [17].

Discursive silences guide inferences and reveal a special orientation towards (or sense of) the communicative act. They are markers of agreement or disagreement, intensifiers or attenuators, by deception or by concealment, argumentative, humorous or ironic. Structured silences are used to distribute communication turns, to mark non-preferred responses, to signal coordination errors, to energise the conversation, to promote changes of topic or to request attention or support [17]. Other silences may have emotional or expressive functions, sometimes acting as a visceral response (or emotional restraint) in reaction to others. These silences are intended to mentally design or prepare the discourse, which is why Camargo and Méndez (2014) [17] include them among “epistemological and psychological silences”, performing cognitive or reflective functions associated with caution, emotion, transgression or resistance. Finally, normative silences obey prevailing conventions in the social groups to which the informants belong. These silences are based on the system of rules, norms, taboos and rituals currently applied in the society considered. These silences, therefore, are determined by the informant’s society, situation and culture. They are formally constructed, and their knowledge and correct use are essential to successful communicative exchanges and to preserving the image and social relationships of the participants. These conventions are based on the principles and values accepted and practised by the speakers in their daily interactions, who consider them appropriate, relevant and fully incorporated into their specific socio-cultural context. Among others, these types of silences perform functions related to situational, social and cultural conventions (Camargo and Méndez, 2014) [17].

The aim of the present study is to consider the functions of the psychological and normative silences [17] presented by pregnant and breastfeeding women living in two regions of Spain regarding their perceptions of environmental risks and food safety [18,19,20,21,22]. There is growing evidence that continued exposure to low doses of pollutants, as occurs in most of the general population, could increase the risk of developing various chronic pathologies, such as cancer, cardiovascular disease or diabetes [20,23,24,25], and also that exposure to various persistent toxic compounds (Polychlorinated Biphenyls, DDT, Heptachlor, Hexachlorobenzene, Dioxins…) during pregnancy has been associated with alterations in foetal growth, new-born weight, asthma, risk of pre-term birth or alterations in neurodevelopment, among others. Persistent toxic compounds are chemicals used in agricultural and industrial production that accumulate in the body in small doses, mainly through the consumption of foods containing animal fats, and carry a silent risk due to the short-term invisibility of their consequences, as well as a diffuse and multifactorial causation of diseases and disorders resulting from this type of toxicity [26,27]. Internal contamination of human bodies by these chemicals is the result of systemic processes involving exposure, absorption and accumulation of compounds, usually in organs and fatty tissues.

Specifically, the silences interpreted in this study are those occurring within the narratives offered in interviews, focus groups and ethnographies of the participating women, resident either in Almanzora (Andalusia, SE Spain) or Tarragona (Catalonia, NE Spain).

To provide some context for our research, these areas were selected for study due to their specific, relevant circumstances. In January 1966, in Palomares, very near Almanzora, two US military planes, one of which was carrying four atomic bombs, collided during aerial refuelling and crashed. Although more than fifty years have passed since then, the nine kilograms of plutonium spilled have not been completely cleared, so the area is still potentially contaminated by plutonium and americium [28,29].

The second study area is Tarragona, a city with an important petrochemical industry, which generates abundant waste, making the local air among the most polluted in the whole country [30,31,32,33]. The city and its skyline are defined by this industry and by the widespread industrialisation that began in the early 1960s, affecting not only Tarragona but also nearby towns, including Vila Seca, Reus, Contantí, La Pobla de Mafumet and El Morell, all of which form part of the context of our field work. Local attitudes towards the chemical industry are ambivalent; its dangers are recognised, but significant economic benefits are also generated. In the two towns (Palomares and Tarragona), pollution and environmental degradation can have a negative impact on public health. For this reason, governments legislate to protect their citizens from these potential sources of harm. In contrast, however, they sometimes minimise the potential health risks of environmental and food contaminants, whether motivated by ignorance, by lack of scientific evidence or by the wish to avoid generating social alarm. In Palomares and Tarragona, environmental risks have been minimized for decades through the biopolitical practices previously described by Michael Foucault [34].

Specifically, in this study, we interpret the psychological and normative silences detected in the discourse of pregnant and breastfeeding women regarding their assessment of the potential dangers arising from dietary and environmental contamination. In addition, we explore the emotional and precautionary motivations that may be reflected in these individual silences during discourses on food safety and on processes of resistance to social, cultural and contextual conventions, in the view that these individual expressions may be determinants of collective silences in the assessment of environmental contamination.

## 2. Materials and Methods

### 2.1. Study Design

This study is based on earlier work conducted in two phases, in 2015 (Ref. CSO2014-58144-P) and 2018 (Ref. AP-0139-2017), exploring why and how pregnant and breastfeeding women excluded persistent toxic compounds from their diet. These investigations were descriptive, interpretive and observational and were undertaken under a qualitative research paradigm based on a phenomenological and ethnographic perspective [35].

The study was designed taking into account previous recommendations for qualitative research [36]. Specifically, our investigation of the meanings underlying discourse and silences is intended to be flexible and inductive, working from the particular to the general [37], seeking to determine the perceptions and demands of the study participants regarding the risks of dietary and environmental contamination, in the context of their place of residence.

### 2.2. Study Sample, Participants and Context

The health centres and the women addressed in this research were selected by intentional (or rational) non-probability sampling [38], taking an equitable, non-discriminatory approach. The following inclusion criteria were applied: women born in Spain, 20 weeks pregnant or more, or who had given birth during the last six months and were breastfeeding (exclusively or also using formula). Socioeconomic diversity was also sought (Table 1).

The field work was carried out at six primary care health centres in two Spanish towns: Tarragona (in Catalonia, in northern Spain) and Valle del Almanzora (in Andalusia, SE Spain) (Figure 1).

The fieldwork for the first study began in January 2016, once approval had been obtained from the corresponding ethics committees in Catalonia and Andalusia, and ended in September of the same year. All participants were informed of the objectives and methods of the research, and written informed consent was obtained in every case. The three ethnographies were conducted from May 2016 to September 2017 in the two regions: the first in Tarragona city centre and in one of the neighbourhoods. the second in two municipalities in the Ribera d’Ebre area (also in Catalonia) and the third in two municipalities in Valle del Almanzora.

The study population was composed of 88 women, of whom 46 were pregnant and 42 were breastfeeding. All were given detailed, extensive information about the study before the interviews, focus groups and ethnographies were held, both in written form and in a meeting with one of the researchers. The informed consent form was separate from the research information document. The protocols for the two studies were approved by three ethics committees: “Parc de Salut Mar” Clinical Research Ethics Committee (Ref: 2015/6459/I), IDIAP Jordi Gol Ethics Committee for Clinical Research (Ref. P15/135) and the Research Ethics Committee of the Andalusian Health Service in Almería (Ref: 66/2017).

The two investigations were conducted in accordance with the Declaration of Helsinki [39].

### 2.3. Data Collection and Field Work Instruments

The methodological instruments employed to obtain the study data were 50 semi-structured interviews with the participating women, three focused ethnographies and five focus groups.

The semi-structured interviews conducted helped ensure the discourse data obtained were of sufficient density and complexity for meaningful analysis [40,41]. Each interview took the form of five sections, addressing the following topics: trust/distrust of food, origin of the food consumed, food preparation, where and with whom food was eaten and knowledge about persistent compound toxins in food. These interviews revealed how the women perceive their relationships with food and the environment, as well as the value systems and norms that underpin these practices [35,42,43].

The focus groups helped promote interaction among the women, enabling them to offer first-hand information. Participation in these groups encouraged discussion and facilitated an open, flexible discourse characterised by intersubjectivity and reflexivity. Observation and analysis of the focus groups also helped the researchers identify and interpret nonverbal behaviours more comprehensively and to contextualise the silences produced [37].

The field notes and in-depth observations compiled during the three ethnographies completed our set of methodological instruments, enabling us to triangulate methods and data and thus effectively compare the participants’ expressions regarding their preferences, norms, social representations, and food and health practices within different research contexts. This comparative approach also allowed us to analyse how medical norms and recommendations on environmental and nutritional risks are accepted, modified or ignored by the women in our study group [37].

### 2.4. Data Categorization and Analysis

Previous categories were determined to codify the narratives and facilitate their subsequent analysis. The categorization was carried out following to the classification of silences according to their pragmatic functions described by Camargo and Méndez [2,14,15] in Spanish society from the analysis of situational contexts. In this way, a dynamic categorization of silence was carried out based on Prototype Theory [16] as indicated above. Silences in speech when addressing risk-related issues were recorded in the field notes and later highlighted in the transcripts of the interviews and focus groups. The authors of this article were involved in every phase of the process: selection of participants, observation scenarios, implementation of techniques, recording, organization of the information and analysis. Together, we built a framework that allowed us to analyse not only the discursive aspects but also the silences, understood as a communicative act that requires interpretation. Silences were coded into two main categories: epistemological and psychological silences and normative silences. The category of epistemological and psychological silences groups four categories of silences: cognitive, precautionary, emotional and transgressive. Additionally, the category of normative silences represents three types of silences: silence as a situational convention, silence as a social conventions and silence as a cultural conventions.

The women’s discourses were digitally recorded during the interviews and focus groups. During the latter, in addition, two of the researchers took notes and observed the participants’ non-verbal language, while a third researcher moderated the proceedings. Interviews were conducted during the ethnographic work; the researchers also took field notes that were later digitized for computer processing and analysis according to the previous categories.

The interviews and group sessions were professionally transcribed and the discourses were encoded using N-Vivo 12 software.

Following the analytical schemes for qualitative research recommended by Atkinson and Hammersley (1994), Ruíz (2003) and Flick (2007) [35,42,43], our analysis highlighted the reality of the women’s discourses, clarified the relationships among these discourses and synthesised the data into a structured whole. The encoding process synthesised and grouped these data, creating an appropriate analytical category for each discursive topic. In the transcripts, the women’s discursive silences are represented by three dots in parentheses, i.e., “(…)”. The names of the women that appears in the narratives shown in the results are not real. The women have been assigned nicknames to maintain anonymity.

The above procedure provided the basis for an interpretive analysis of discourse and silences from a hermeneutical perspective [44]. This global vision of the results obtained helped us correlate the values and beliefs emanating from the discourses with various social theories.

## 3. Results

### 3.1. This Epistemological and Psychological Silences

This first section presents the women’s spoken narratives, together with silences reflecting affective, expressive and cognitive dimensions of their discourse. In the context of our field work, these silences can be interpreted as emotional restraints or acts of hesitation during the participants’ reflection. Epistemological and psychological silences imply the representation of potential pragmatic functions, i.e., those related to cognition, reflection, caution, emotion, transgression or resistance.

#### 3.1.1. Cognitive Silence

The women taking part in our study remained silent and reflective, doubting what exactly they wished to say. This silence allowed them to mentally organise their discourse about environmental risks in their surroundings.

-Moderator:And in Palomares, do you work in a food-related area?

-Participant:In a (…) packaging warehouse.

-Moderator:In Palomares, do people talk about this issue naturally or do they keep silent?

-Participant:No. They don’t speak about it naturally, but they don’t try to hide it either. If the topic comes up in the conversation, they speak with total frankness, but they say they have absolutely no worries. We still haven’t heard of any case specifically related to that, to that question. Yes, there may be, (…) it could happen anywhere.

(Tatiana, pregnant woman, age 36, Vera focus group)

In this account, the participant pauses twice to reflect on what she wishes to say regarding her work and the risk of contamination in Palomares.

-Moderator:Do you think that in your environment, where you all live, there is any biological, chemical or radioactive risk to your health?

-Participant:We are in Tarragona, I mean (…). We don’t have three eyes, because we haven’t mutated yet, but (…)

-Moderator:Are you all quite sure of that?

-Participant:Yes.

(Lucía, breastfeeding mother, age 30, “Tarragona II” focus group)

On this occasion, the participant from Tarragona hesitates, giving herself time to formulate her perception of risk via a metaphorical remark. 

#### 3.1.2. Precautionary Silence

On some occasions, the participants appeared cautious or repressed their opinions.

“Those in charge tell us (…) what they want. If you like, you believe it. And if you don’t, you don’t believe it. But you don’t do anything either. So, in the end (…)”(Isabel, pregnant woman, age 37, Tarragona I focus group)

In this participant’s discourse, there were silences, holding back her opinion on the possible negligence of the institutions in protecting the environment in and around Tarragona and on the failure of civil society to react to this situation.

-Moderator:Don’t the people of Palomares talk about the nuclear accident?

-Participant:Yes, yes, of course. They do talk. We do, me and my friends (…). Yes, we said it might have had an effect, too. We said that this was a bit contaminated and (…). But I’d say (…). I’d say, if it was contaminated there wouldn’t be people living here. What I mean is, for example, when the plants grow (…)

-Moderator:So, do you think that around Palomares there is a possibility of contamination now?

-Participant:Well, I think that when it happened (…). I don’t know. I don’t know.

-Participant:The thing is, maybe, in my case, I think that things like pesticides (…), maybe they do more harm to the countryside or whatever, than the radioactivity.

(Sofía, breastfeeding mother, age 28, Vera focus group)

In this example, the participant was silent at various times, holding back an opinion that might have made her uncomfortable.

#### 3.1.3. Emotional Silence

During their discourse, some participants were flustered and seemed emotional when talking about environmental pollution in their community, and these circumstances made it impossible for them to continue in words.

“For example, I eat a lot of fresh fish, my father goes out fishing, and you can’t even trust that, because you say (…) in the water (…) the spills from the ships (…). Perhaps when it all happened there was more perception of risk, but as time went by, the thing is (…) people don’t pay any attention, it’s just something that happened in the past, it’s all forgotten now, nobody talks about it. It happened and that’s that”(Juana, pregnant woman, age 32, Vera focus group)

This participant became emotional as she tried to describe how people in Palomares today set little store in the risk of contamination from plutonium and americium.

“For me, it has a direct influence, because the plants, from the rain, from the sky, and with the land contaminated, depend on it. And the animals that eat contaminated grass, or that eat, well, everything (…) everything, everything, nitrates, everything that’s in the soil, and then everything goes into the plants. And we eat all of that (…) so yes, I do think it affects us. And here, in Tarragona, especially. What’s more, we don’t know what we’re breathing (…) I think it affects us a lot”(Verónica, breastfeeding mother, age 39, Tarragona I focus group)

This woman, too, was unhappy and showed her distress at the potential health risks from environmental pollution in Tarragona.

#### 3.1.4. Transgressive Silence

At times, the participants fell silent during their discourse as a sign of disagreement or rebellion in response to an argument that was mostly accepted by the group or expressed by the researchers.

-Moderator:Do you grow your own fruit and vegetables?

-Participant 1:Yes, because organic produce is really expensive at the shops.

-Participant 2:I used to grow things in my garden, but I stopped (…), because (…) I’d say, “I live here, in Ramón y Cajal, and I think I’m eating less healthy food than if I was shopping at the greengrocer’s”. Because the cars are going past all day. And, in the end, if I am sweeping up the dirt and it all comes up black because of the pollution, then that same black stuff has been taken up by the plants. And that’s why I stopped, in fact. You can’t trust the environment.

(Magdalena, Participant 1, breastfeeding mother, age 37, Tarragona I focus group)

(María, Participant 2, pregnant woman, age 32, Tarragona I focus group)

Participant 2 started speaking and then paused, implying her disagreement about the supposed benefits of food from urban gardens in Tarragona. She found it hard to believe that an urban garden there minimises the risk of food contamination.

-Moderator:But there are controls on environmental pollution.

-Participant:Yes, that’s what it seems (…) If it is controlled, sometimes (…) but the food industry will be manipulating the system, they do their own analyses (…) and you can also see how they’ve got around the controls. It is controlled, yes, I’m sure, I’ve seen it, I’ve seen food plants fitted (…) with waste controls. Everything is controlled, but there’s always someone who gets around it, because what comes first is always the economic interest.

(Ana, breastfeeding mother, age 30, interview, Cuevas de Almanzora)

This participant stopped speaking on several occasions during her interviews when the question of administrative control over environmental pollution was addressed, appearing reluctant to believe that the controls are effective—she works for a food company near Palomares.

### 3.2. Normative Silence

Normative silences are governed by situational, social and cultural conventions, which impose a body of norms and rules that a speaking community accepts and practices to ensure harmonious interactions.

#### 3.2.1. Silence as a Situational Convention

Various participants fell silent during their discourse in response to the fieldwork context and format of the situation, especially during the focus groups.

“In fact, I don’t know a lot of people there, but what I’m saying is that anecdotally, when we don’t agree on something, I tell them, “That’s because you’re from Palomares” (…) but no (…) I don’t think it’s that they have worse health than us living in Vera, or those who live in (…) or those who live in (…). Health-wise, they’re all much the same”(Irina, pregnant woman, age 34, Vera focus group)

The participant paused after joking about the relationship between the non-perception of contamination risks in Palomares and the potential stigma attached to the population. She then sought to justify her ironic comment and fell silent again, realising that in the current situation and context (the discussion within the focus group) it was better to remain silent.

-Moderator:And do you think that the environment also affects food, can it affect your food and your health?

-Participant:Of course. Just go there, look at the refinery, see the trees that aren’t even green, the leaves are brown. Even so (…), we’ve eaten vegetables and things grown here and, for now, we’re OK. Touch wood, right? But a little, I think, it must have an effect, like everything else. But neither do I think there’s much difference between living here and living in the centre of Barcelona. Everywhere you go (…).

(Paula, breastfeeding mother, age 27, El Morell focus group)

In this example, a woman from Tarragona, after remarking that contamination is visible in the form of soot impregnated in the trees, then fell silent so as not to be so explicit, and even sought to minimise the impact of her comment. She understood that the focus group is not the right situation in which to talk about these matters, as it might cause discomfort and fear in other women who were pregnant or feeding their babies.

#### 3.2.2. Silence Due to Social Conventions

Some silences arose from the asymmetric relationship between the participants and the moderators/researchers, who were viewed as experts in health and environmental issues.

-Participant:Then, they also blamed (…) with so much internet, they said there was an array of solar panels for internet. They blamed the panels more than (…) than that, than contamination, which was a lot.

-Moderator:Do people in Palomares think that these foods could be radioactively contaminated?

-Participant:Well, I don’t know about that. But where I was, in the company where I worked (…) they said (…) that yes, that that could also have an effect. That it was a bit contaminated by (…). But I said (…) I was thinking, if it was contaminated, there wouldn’t be anybody living here.

(Andrea, pregnant woman, age 37, Tíjola focus group)

This participant had certain qualms about expressing her opinion. In the social context in which the focus group took place, she may have felt she was exposed to judgment. She also may have considered that in this context it was not appropriate to relate private conversations with her workmates about the environmental risks in Palomares.

-Moderator:And to finish, the last section is the issue of chemical substances in food. Whether you think that food contains these substances.

-Participant:Yes, but I don’t know to what extent (…), I’m totally unaware, that is, at what level, what effect it might have, do they remain or do they disappear? I don’t know what to say (…)

-Moderator:To simplify, do you think these chemical substances accumulate or not?

-Participant:I believe that if you (…) They don’t disappear if you regularly eat stuff that contains it (…), isn’t that right?

-Moderator:In other words, for you, they do accumulate, you’d say.

-Participant:They accumulate because you don’t have time to clean them out if you’re always eating products that contain them. But if you don’t eat them, then yes, they will, because your body is healthy. But if you eat, if you eat them (…)

-Moderator:You accumulate them, over time.

-Participant:I don’t know if accumulate is the word! Or… Yes, you do. I can’t explain scientifically how it works. The truth is that (…) I really don’t know.

(Elena, pregnant woman, age 20, Tarragona I focus group)

This participant was insecure and fell silent several times during her intervention. Her speech was not spontaneous, and she excused her ignorance on several occasions, probably because she felt she was being judged in the social context of the focus group.

#### 3.2.3. Silence Due to Cultural Conventions

The participants’ discourse frequently features culturally motivated silences, expressing local beliefs and taboos about the potential risks of environmental contamination in their neighbourhood.

-Moderator:Do you believe that environmental pollution affects food?

-Participant:Yes, I’m sure it does (…). It would be the same as the food. In other words, at the legal level, everything is controlled, but really, they could be doing anything, which isn’t (…)

(Cristina, breastfeeding mother, age 35, Tarragona II focus group)

The moderator asked about one of the “taboo” topics in the Tarragona area, namely the potential contamination of food grown in the area due to environmental pollution. After she started speaking, the participant first paused after saying that environmental contamination affects food. She then mentioned the culturally motivated lack of confidence among people in Tarragona regarding the levels of contamination and its control by the authorities.

-Moderator:And what do people think in Palomares about the possibility of potential contamination from the nuclear accident?

-Participant:They [the people who live in Palomares] (…), well (…), they are totally confident, they say they’re planting their crops, they say it’s been many years since it all happened, they haven’t seen any diseases related to it, or paid much attention to it or heard anything alarming. They’re totally confident, they regularly eat local products.

(Eva, breastfeeding mother, age 29, Vera focus group)

The woman remained silent at first and then reproduced the official discourse among the inhabitants of Palomares that has been culturally established since the beginning in dealing with the nuclear accident.

## 4. Discussion

This study shows that psychological and epistemic silences can be highly significant, indicating the cognitive and emotional state of the participants in a communicative interaction. The social orientation of our analysis also highlights the importance of normative silences, i.e., those that are motivated by situational, social and cultural conventions. As observed by Méndez (2016) [45], silence is an element that is present in communication and has meaning and is used to express meaning. From this idea, we can deduce its main communicative purpose, that of transmitting information. The psychological and normative silences [17] produced by the participants in these interviews, focus groups and ethnographies constitute an important source of information. The contextualisation of this information in the home towns of the women taking part enables us to draw inferences about their perceptions of environmental dangers and the risk of food contamination.

Our discourse analysis reveals evidence of the denial of environmental and food risk by pregnant and breastfeeding women who live in two highly contaminated areas of Spain. The first area, Tarragona in NE Spain, is contaminated by chemical emissions, while the second, Palomares in SE Spain, continues to be affected by radioactive compounds (plutonium and its decay into americium).

According to our interpretive analysis, the types of silence most commonly presented by these participants were psychological or epistemic [17], reflecting the women’s emotions, feelings, hesitations and degree of interest in talking about the impact of contamination in their home towns. During their interventions, the women made numerous silences, which corresponded to reflective or doubtful attitudes and allowed them to mentally organise their speech and avoid saying anything not considered socially correct. This circumstance was more evident in the focus groups, probably because the participants did not know each other [5]. This would also have increased their caution in expressing perceptions of risk related to environmental contamination. In the two areas where our field work was conducted, this question has been silenced for many years. In Tarragona, local chemical companies are major employers, which leads the population to normalise the situation and to downplay the risks of living in an area surrounded by chimneys and industrial activity. Palomares, meanwhile, was the site of an aviation accident that took place on 17 January 1966, causing four nuclear bombs to fall to earth. They did not explode, but two broke up on impact [46]. From the outset, the government (at the time, a dictatorship) covered up the consequences of this nuclear accident due to cold war geopolitical and military considerations [47]. The local inhabitants, too, have always minimised the impact of the accident on the environment. Moreover, and unlike in Tarragona, no social benefit was derived nor was any identarian movement generated. In Palomares, the sense of belonging, people’s roots and their sense of identity were precisely the qualities that the inhabitants did not want to lose; their goal was to continue being a population of fishermen and farmers as always and not a society marked by nuclear radiation. This was perfectly illustrated by one participant when she said, “People don’t pay any attention, it’s just something that happened in the past, it’s all forgotten now, nobody talks about it. It happened and that’s that”.

Although in both study areas our participants discounted the possible impact of contamination, concerns were expressed about potential health risks, about inadequate administrative controls on environmental pollution and about the possible transfer of contamination to the local foods consumed. Previous work by our research group has shown that, in general, pregnant and breastfeeding women place greater trust in homegrown foods (cultivated in gardens or allotments) or foods purchased from small businesses that source their products from local farmers [18,19,48]. However, this preference was less clear among the women living in the two study areas affected by contamination. In the days following the nuclear accident, all the crops in and around Palomares, mainly tomato plantations, were uprooted, and the fishing fleet in Villaricos (a coastal town near Palomares) was not allowed to sail, since the fourth bomb fell into the sea and was not found until 79 days later [47]. However, as the years passed, and with governmental propaganda asserting the absence of contamination, the agricultural industry again flourished in the region. Nevertheless, in nearby towns there remains reluctance to consume foods labelled as having been produced in Palomares or fish captured in local waters. In Tarragona, too, the fear of environmental contamination in local food generates distrust. Some women living in this area were even sceptical of the benefit of growing vegetables in their own organic gardens, insisting that their ecological practices were irrelevant if the land was already contaminated by more than a century of factory discharges into the environment.

The women who took part in our study repeatedly stated that potential food contamination related to environmental degradation was unavoidable. This outlook was corroborated by the psychological and epistemic silences produced during their interviews and focus group interventions. Thus, several participants paused significantly during their discourses, after which they continued, “we couldn’t eat anything” in reference to the quality of the local food or “we’d have to live in a bubble”, remarking on the impact of environmental pollution. In other words, these participants acknowledge that they are unable to alter their diet and lifestyle and are resigned to accepting the situation and not attempting to change their reality. This pattern of behaviour is what Seligman [49] called learned helplessness, or the reaction of giving up based on the belief that nothing can be done to change the situation [50]. The participants in our study believe that it is impossible to defend themselves against potential food or environmental contamination, and so they present behaviours that are characteristic of learned helplessness. According to Seligman (1991), this behaviour is typical of persons who are psychologically inclined not to defend themselves against perceived risks. This attitude may have arisen because at some time they attempted to defend themselves but were unsuccessful; faced with a constant negative outcome, they then adopted an attitude of non-response, believing that no change would occur in any case. This idea is very clearly expressed in the women’s narratives, when they affirm that, by their own actions, there is no way of changing their diet and lifestyle.

Another aspect examined in our analysis is that of normative silences [17], when participants’ speech was inhibited by the context of the interaction. In our study, the fact that the field work took place near the women’s places of residence led them to remain silent at certain moments during their discourse. Similarly, the communicative situation generated in the focus groups favoured silences to a greater extent than in the individual interviews and ethnographies. In this context, on certain occasions, social conventions also favoured silences, due to asymmetric relationships among the women regarding knowledge of the environmental and food risks in their home towns. However, the most frequent normative silences were those related to cultural motives and “taboo subjects” [51], such as the traditional minimisation of environmental risks in and around Palomares and Tarragona. In these two communities, public opinion generated a “spiral of silence” [8] around the contamination of the environment. Following Noelle-Neuman (2003) [8], we believe that the media played a decisive role in swaying public opinion in the two towns, minimising perceptions of environmental risk. In the particular case of Palomares, the plane crash that took place on Monday, 17 January 1966, marked the beginning of Spain’s planning and control of its image of Spain in the field of crisis management. According to Micaletto (2016) [52], the first step taken by the Spanish government to manage this crisis was to employ the media to deliver its messages. Specifically, the press merely reported that the two planes had collided whilst silencing all other circumstances. Thus, the ramifications of the incident were largely silenced, and the domestic population remained unaware of its reasons, the possible consequences and the actions taken regarding sanitary and legal protocols. For decades, this silence was largely maintained by the different healthcare and technical institutions responsible for monitoring the health of the population and for measuring levels of plutonium and americium in the surroundings of Palomares. In fact, only one, limited, radiation control project was undertaken among the local population (termed the Indalo Project), aimed at dosimetrically studying the radiation present in 1077 inhabitants of Palomares, but not the effects of plutonium on their health. The results were not made public or provided to the affected population. The official justification for this is that ionising radiation has not been related to any type of disease. From the outset, much of the research conducted in this area has been shrouded in secrecy, and no reliable epidemiological study has yet been conducted in the area [29]. However, in the USA, health authorities monitored the 1600 military personnel who performed clean-up operations in the area after the accident, potentially exposing themselves to contamination by plutonium. For many years, the Americans claimed that this exposure was not sufficient to make the soldiers sick. This continued until 2013, when it was recognised that the plutonium levels recorded were high enough to suggest a connection between the exposure in Palomares and the lung and bone cancers suffered by some veterans, as well as the liver cancers diagnosed before 1990 and the leukaemia diagnosed before 1982 [28].

In Tarragona, the normative silences detected in the narratives of the participants are culturally and socially based on prevailing economic interests, as the petrochemical industry is the main employer and source of financial wealth in the area. The petrochemical complex is estimated to account for around 30,000 jobs, directly and indirectly [53]. Accordingly, the topic of contamination was locally taboo for many years, and residents were initially suspicious when initiatives were taken to acknowledge and study the level of environmental contamination in the area [31]. The chemical industry was established in the province of Tarragona at the end of the 19th century. Until the middle of the 20th century, its development was uneven, due to domestic and international political instability, but since 1945, this industry has grown steadily and the Tarragona complex is now the most important in southern Europe [54]. Although for decades the population witnessed smoke from chimneys and waste generated from factories, it was not until the beginning of the 21st century that mobilisation began and demands were made for this contamination to be controlled [55]. Disinformation and a strong sense of local identity generated a “spiral of silence” around the question of environmental contamination in Tarragona. As observed by Grijelmo (2012) [56], the genesis of this social silence was the information given out (or more exactly, the disinformation and/or absence of information) and the fact that institutions often determined, more or less subtly, which information was made public and which was withheld.

In our view, both in Tarragona and in Palomares, the authorities have failed to inform the population properly about the potential dangers to health from environmental contamination; instead, they have fostered silence and passivity among civil society. According to Foucault (2007) [34], this approach to public health is a typical aspect of the exercise of biopolitics and biopower by institutions. The operability of biopower includes a series of characteristic elements, all of which we observed in the two study areas. Firstly, the state institutions emit a discourse of “truth”, minimising the environmental risks that may be present. This discourse is then reproduced throughout society via collective intervention strategies. Finally, modes of subjectivation are used as a formative practice of identity processes [57]. According to Foucault (2007) [34], biopower induces subjectivity in individuals, through processes that attenuate their reflective capacity on health issues. When this biopower is exercised by public institutions, as biopolitics, a new element, “the population”, is established as a biological problem. Biopolitics characteristically addresses collective phenomena, including their economic and political effects; in other words, phenomena that are individually random and unpredictable, but for which at a collective level certain constants can be determined [58], for example, the risks to health posed by environmental contamination. 

## 5. Conclusions

Industrialisation and modern lifestyles are known to produce adverse effects on the environment. In turn, pollution and environmental degradation can have a negative impact on public health. For this reason, governments legislate to protect their citizens from these potential sources of harm. In contrast, however, they sometimes minimise the potential health risks of environmental and food contaminants.

Our study shows that both in Palomares and in Tarragona, discourses of official truth were generated that minimised or denied the existence of environmental risk, creating a “spiral of silence” about the risk to people’s health. Our analysis of the psychological and normative silences produced in the narratives of the study participants shows that most of these women are cautious in their words and minimise the risk of contamination present in their environment, thus reproducing the institutional discourse. Nevertheless, these women are doubtful of the quality of their environment and the food that is produced there.

Despite the evident problems, the populations of Palomares and Tarragona have for many years remained passive, doing little to defend their rights and health. In contrast, the US government recently acknowledged that some of the soldiers who had worked in Palomares during the recovery of the four nuclear bombs and may have been exposed to plutonium had subsequently contracted cancer. On the other hand, none of the cases of local inhabitants who have suffered from cancer in the last 55 years have been linked to plutonium contamination, despite the fact that their potential exposure has been much longer lasting than that of the American soldiers. In Tarragona, although the incidence of cancer there has risen in recent decades and is now the second highest in Spain, to our knowledge no judicial decisions have corroborated any causality between contamination and the presence of cancer among the general population. Recently, however, the High Court of Justice of Catalonia ruled in favour of a worker who fell sick with cancer after being employed in one of the leading chemical companies in Tarragona. This court associated the presence of the disease with the worker’s lack of protective equipment and his consequent exposure to ethyl, benzene and other chemical derivatives.

In conclusion, we believe that the populations considered in this analysis present the phenomenon of “learned helplessness”, a condition in which people believe that no action is possible to reverse the degradation of their environment and the potential contamination of the food they consume. This condition has arisen from practices of biopolitics and biopower long exercised by public institutions, which have generated situations of grave social injustice for the populations in question.

This study provides a valuable and introductory study on precautionary attitudes and minimization of environmental risk of pregnant and lactating women who live in potentially contaminated places. As a limitation, we want to highlight that the influences that the immediate context and social agents (family, friends, teachers, health personnel…) have on the narratives and silences of women have not been studied in depth. In this way, the results emanating from this study should be considered basic material for future research that studies the influence of the close context of women in the interpretation and assessment of environmental and dietary risk during pregnancy and lactation.

## Figures and Tables

**Figure 1 nutrients-14-00593-f001:**
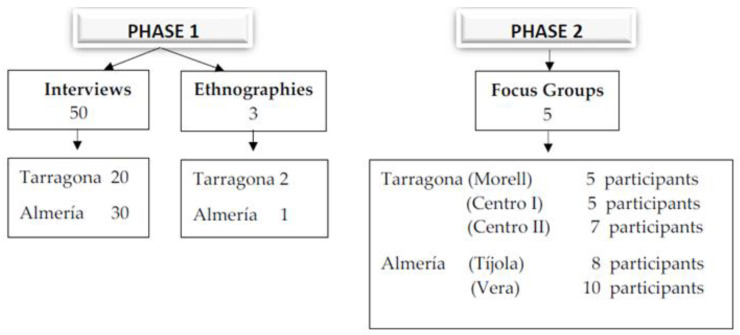
Data collection instruments by field work place.

**Table 1 nutrients-14-00593-t001:** Characteristics of the pregnant and breastfeeding women participants.

	Pregnant	Breastfeeding
Participants		
88	46	42
Age range		
Age 20–29	8	6
Age 30–39	34	32
Age 40+	4	4
Education Level		
Primary	4	3
Secondary	15	17
Higher	27	22
Number of children		
1 child	27	24
2 child	14	13
3 child or +	5	3
Autonomous Community		
Andalucía	28	21
Cataluña	18	21

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
