# Peer review of "The Sound of Silence: Unspoken Meaning in the Discourse of Pregnant and Breastfeeding Women on Environmental Risks and Food Safety in Spain"

_nutrients, 2022, doi:10.3390/nu14030593_

Round 1

Reviewer 1 Report

In this manuscript, the authors utilize transcripts from previously published and analyzed focus groups in order to interpret the silences exhibited by participants. Although this is a unique and interesting study, given that silences and moments of potential discomfort and stress are part of focus group research, it was unclear how the authors categorized the silences. In the methods the authors state that they utilize traditional methodologies for qualitative research. However, these usually utilize dialogue to categorize themes. In this study the authors specifically utilize silences. The methodology needs to be more comprehensive as to how they assessed and categorized the silences into cognitive vs precautionary vs emotional silences etc. Did three interpreters agree on the categorizations? 

Figure 2 (ie the map of Spain) can be removed as it is not necessary for understanding the study. 

The authors mention ethnographies throughout the manuscript. It is unclear how these were utilized in this specific analysis. 

Author Response

Consulte el archivo adjunto

Reviewer 2 Report

Dear Authors

Very interesting study, but you have received a large extension of the introduction section on the meaning of silence, while you mention little about the effects of infected areas on pregnancy and breastfeeding. Also if data on the epidemiology of these effects are available, they should be reported. Furthermore, explain what effect does this infection have on the perinatal period?

Cite articles showing that there is environmental pollution

What are the persistent complex toxins in food?

In the results section, give some nicknames to women

The conclusions section has a substantial problem. In this section, we do not use references; instead, we quote the most important findings of our study and suggest solutions to the problem. What do you suggest about changing the attitude of the population? Suggest information through official organizations, mobilization of the population through organizations, increased public health measures and screening in the special populations.

What are the reasons for this silence of women? Is the lack of information or the lack of education of the population the causes of this situation? Lines 647-661 should be transferred to the introduction because they provide knowledge of the already existing problem.

Strengths and limitations of the study should also be added

Round 2

Reviewer 1 Report

The additional methodology helps to support the categorization of the silences.